# A qualitative analysis of female sex workers' lived experiences with adherence to Pre-exposure Prophylaxis (PrEP) in Zimbabwe

**Definate Nhamo**[1]*, **Dixon Chibanda**[2], **Frances M. Cowan**[3,4], **Sinegugu E. Duma**[1]

1 School of Nursing and Public Health, University of KwaZulu-Natal, Durban, South Africa, 2 Department of Psychiatry, University of Zimbabwe, Harare, Zimbabwe, 3 Centre for Sexual Health and HIV Research (CeSHHAR) Zimbabwe, Harare, Zimbabwe, 4 Department of International Public Health, Liverpool School of Tropical Medicine (LSTM), Liverpool, United Kingdom

* defnh78@gmail.com

## Abstract

### Background

Female sex workers (FSWs) are at an elevated risk of HIV infection with an eight-fold risk of HIV infection. In countries like Zimbabwe, FSWs have an HIV incidence of around 10.2%. With this elevated risk, the World Health Organization has prioritized Female sex workers (FSWS) for PrEP – an HIV prevention option taken as a daily pill during periods of risk but, FSWs continue to experience challenges with daily PrEP adherence due to daily dosing, related side effects, ARV stigma and low risk perception. This article presents the FSWs' lived experiences with PrEP adherence in Zimbabwe.

### Methods

We purposively identified twenty FSWs and conducted individual interviews to understand FSW lived experiences with PrEP adherence. We applied Colaizzi's seven steps of phenomenological analysis to develop the themes.

### Findings

Three main themes emerged, namely positive experiences with PrEP adherence, negative experiences with PrEP adherence and the meaning attached to PrEP adherence. The positive experiences theme had four sub-themes as, overcoming PrEP-related forgetfulness, overcoming mobility-related PrEP disruptions, overcoming COVID-19 pandemic-related PrEP experiences and overcoming PrEP-related side effects. The negative experiences theme had two sub-themes including, enduring GBV and stigma associated with PrEP use and, COVID-19-related disruptions to PrEP adherence. The third emerging them was on the meaning attached to PrEP adherence. This theme had one sub-theme on PrEP adherence as a survival strategy.

**Data availability statement:** All relevant data are within the paper and its Supporting Information files.

**Funding:** The author(s) received no specific funding for this work.

**Competing interests:** The authors have declared that no competing interests exist.

## Conclusion

Whilst FSWs reported both positive and negative experiences with PrEP adherence, it is important that FSWs used the meaning they attached to these experiences to take control of their lives and be more determined to use PrEP adherence for survival and protection from HIV. Based on these findings, we recommend close monitoring and support to promote adherence, minimize PrEP discontinuity and promote positive lived experiences with PrEP adherence.

## Background

Globally, Female sex workers (FSWs) have an eight-fold risk of acquiring HIV compared to the wider adult population [1,2]. With this elevated risk of HIV acquisition, the World Health Organization (WHO) has prioritized FSWs for PrEP, however, PrEP adherence remains problematic [3]. (*Consolidated Guidelines 2016.Pdf*, n.d.). In Zimbabwe, estimates place FSWs' HIV prevalence at 40.2% against a 10.49% prevalence in the general population [1,2]. HIV incidence among the same population groups remains exceptionally high at 4.3% [4].

PrEP is the use of daily oral antiretroviral pills by HIV-negative persons to prevent HIV infection, and is one of the widely available HIV prevention strategies [5–10]. The oral PrEP containing Tenofovir is recommended for use. In Zimbabwe TDF/FTC is preferred, with TDF/3TC as an alternative. However, PrEP adherence remains complex due to the required daily dosing, related side effects, ARV stigma and low risk perception [11]. The World Health Organization recommends adherence support to aid PrEP adherence for those on PrEP [3].

A systematic review on PrEP continuation across diverse population groups in 2018, separately found high rates of PrEP discontinuation within the first six months of initiation. These high rates of discontinuation were recorded in observational studies without adherence interventions, across all population groups in sub-Saharan Africa. The reasons for PrEP discontinuation were reported as side effects, stigma, influence of partners, difficulty in accessing services, and reduced HIV-risk perception [12,13].

There is a small but growing body of literature on FSW experiences with PrEP adherence in Africa and globally. A qualitative study in Zimbabwe by Busza et al, 2020, provided some useful insights on early PrEP introduction and continuation among FSWs and highlighted important challenges related to negative influences from family members against PrEP-use. Female sex workers were faced with negative family pressures of discouragement, skepticism, and fear of being given experimental drugs with no proven efficacy. Female sex workers reportedly found it difficult to take oral PrEP as prescribed. There was fear of side effects and possible rare adverse complications as described in study information leaflets [14]. The South African Treatment and Prevention for Sex Workers (TAPS) study highlighted the distrust of PrEP existence and its related efficacy, which in turn affected FSW's motivation and continued use of PrEP [15]. The ACCESS study conducted by Pillay

et al, (2020), provided useful insights in early oral PrEP introduction among FSWs and MSM finding that PrEP information, education and Communication (IEC) materials and perceptions of being at high risk of HIV infection prompted FSWs and MSM to initiate PrEP while side effects and feelings of being stigmatized led some to discontinue [16].

Another qualitative study conducted among South African FSWs by Makhakhe et al, (2022), confirmed how little distinction had been made between PrEP and ART, leading to the stigmatization of FSWs on PrEP. The lack of differentiation reportedly led some clients of SWs to believe that PrEP and ART are the same [17]. In Tanzania, a study conducted by Mantsios et al, (2022) highlighted oral PrEP challenges among FSWs as, problematic daily adherence, ARV stigma, side effects, alcohol use, lack of safe PrEP storage places when in someone's space and fear of violence [18]. Again, in India, Sahay et al, (2021) found similar PrEP challenges among FSWs citing alcohol use, pill fatigue and stigma associated with ARV use as potential challenges to PrEP adherence [19]. All these studies recommended community education on PrEP with correct and consistent messaging around PrEP and making clear distinctions between PrEP and ART. PrEP information was also recommended to be peer-led, whilst addressing both structural and psychosocial constraints to promote safe working environments for FSWs. In light of these early PrEP experiences and related challenges among FSWs, this article aims to highlight the FSWs' psychosocial related experiences with PrEP adherence which could be used to inform the development of a psychosocial intervention to enhance continuation among FSWs in Harare, Zimbabwe.

## Methods

### Setting

The study was conducted in Zimbabwe, a sub-Saharan African country, with a population of about 16.6 million, and a high HIV prevalence of about 10.5%. This study was conducted in Harare, the country's capital city, within a specialized clinic for key populations, including FSWs. The clinic offers comprehensive sexual reproductive health (SRH) services, including provision of oral PrEP to FSWs from all over Harare.

### Study design

We used a qualitative research design within the phenomenological approach to explore and analyze the FSW's lived experiences of PrEP adherence and the meaning attached to it. This was the most suitable approach to use because of its ability to decode the participants' significant life experiences and offer insights into how they make sense of their lived experiences [20]. These qualitative interviews were conducted once-off, with no repeat interviews held.

### Population and sampling technique

The study population consisted of all FSWs who were initiated on oral PrEP within a larger sexual reproductive health and HIV prevention programme in Harare, Zimbabwe. To be eligible, one had to be 18 years or older, identify as a female sex worker, or be on oral PrEP. Purposive sampling technique was used to select and recruit 20 women who self-identified as FSWs and had already been initiated on oral PrEP for at least three months; resided in and around the study catchment area of Harare and were planning to stay there for the subsequent six months to enable any study follow-up visits. This predetermined sample size of 20 FSWs was deemed adequate to elicit an in-depth understanding of the FSW's experiences with PrEP adherence. Small sample sizes are considered adequate in phenomenological studies where the focus is on the richness of data collected from each participant rather than generalizing the findings [21].

### Recruitment process

Recruitment was conducted from 17 June 2020 through to 10 July 2020. The female sex workers presenting at the clinic for their first PrEP follow-up visit were informed about the study, and invited the FSWs to participate in the study. The potential participants were given the opportunity to refuse to participate. However, once the study was explained

thoroughly, all participants who met the criteria were recruited until data saturation was reached. Those interested in participating in the study were given the information sheet and informed consent forms to read, understand, and sign, indicating their voluntary participation in the study, and to return to the clinic for interviews on the second day. The FSWs were reimbursed US$5 to compensate them for their transport costs as recommended by the Medical research council [22].

## Testing of data collection tools

Cognitive interviews were conducted with three FSWs to check for clarity on the semi-structured interview guide questions and whether the questions could be easily understood by the participants. The three FSWs were recruited from the same research setting, met the same eligibility criteria as the main study participants and used the same semi-structured interview guide designed for the main study. The three FSWs confirmed the clarity and comprehension of the questions in soliciting the FSWs' experiences on PrEP adherence [23].

## Data collection

Data was collected between 17 June 2020 and 10 December 2020 which was during the peak of the COVID-19 pandemic. Therefore, all COVID-19 pandemic-related protocols as specified by the University of KwaZulu-Natal and the Zimbabwean Ministry of Health and Child Care, including social distancing and wearing of masks, were observed and applied from recruitment throughout the data collection process.

Detailed notes were taken during the interview. In addition to the notes, all interviews were audio-recorded to ensure accuracy in data capturing. A semi-structured interview guide designed in line with the study objectives, was used to guide the interviews. The questions asked included the FSW's experiences of the use of PrEP and adherence to PrEP. PrEP adherence was defined as the taking PrEP during periods of risk. As these were FSWs, the assumption was that their risk was continuous. Probing questions were asked to elicit a deeper understanding and get more nuanced descriptions of participant's lived experiences with PrEP adherence. The semi-structured interview guide was translated from English into Shona and Ndebele by a professional bi-lingual translator who was conversant with both local languages. Each translated interview guide was then translated back into English to check for consistency and to ensure the original meaning was retained in translation. The translated interview guide was then checked by the first author for any loss in meaning before using them.

Individual face-to-face interviews were conducted with each participant in a private and quiet space within the clinic. Interviews were either conducted in English, Ndebele or Shona, depending on the participants' language preference. Each interview lasted for at least an hour and was audio-recorded. The interviews were transcribed within 24 hours, to ensure the researcher still remembered the interview proceedings and was able to fill in the gaps. All audio files, transcriptions and translated files were kept in a password-protected folder on the researcher's laptop as well and backed up on an external hard drive, in line with both the Biomedical Research Ethics Committee (BREC) and Medical Research Council of Zimbabwe (MRCZ) policies.

## Data analysis

The seven steps of phenomenological data analysis as outlined by Colaizzi, (1978) were applied [21,24]. Firstly, the researcher familiarized herself with the data by reading through all the participant accounts several times. The reading was both by case and across cases in search of meaning. Secondly, all statements in the accounts that were of direct relevance to PrEP adherence were identified. This was done by identifying accounts that made a direct reference to factors that may have affected PrEP adherence among FSWs. Thirdly, The researcher identified meanings relevant to PrEP adherence that arose from a careful consideration of the significant statements. The researcher reflexively "bracketed" her pre-suppositions to stick closely to PrEP adherence experienced (though Colaizzi recognizes that complete bracketing is never possible). Fourthly, clustering of the identified meanings into themes that were common across FSW

PrEP experiences was done. Again bracketing of pre-suppositions was crucial, especially to avoid any potential influence of existing theory and concentrate on the recounted lived PrEP experiences. As a fifth step, The researcher wrote a full and inclusive description of the phenomenon, incorporating all the themes produced at step 4. Sixth, condensing the exhaustive description down to a short, dense statement that captured just those aspects deemed to be essential to the structure of the phenomenon was done, in this case, PrEP adherence. All the coding was done manually by two researchers, the researcher and a fellow qualitative researcher, using an excel sheet. The supervisor, who is a qualitative expert, provided oversight of the coding and analysis process. Lastly, the researcher returned to the FSWs with key findings to ask whether it captured the FSW experiences of PrEP adherence. This process is referred to as member-checking [25]. The researcher would go back and modify earlier steps in the analysis in the light of feedback provided by the FSWs. The findings were informed by a feminist lens.

### Scientific rigor

To ensure credibility, the raw data and the emerging themes were shared with a senior researcher, who is an expert in qualitative research (last author). This was to confirm if the themes developed were accurately informed by the FSWs' voices on their lived experiences. This was an iterative process, characterized by a lot of questions to try and find meaning in the data shared. We conducted member checking, a process to determine the accuracy of themes and interpretation of data through taking back the themes to participants and determining whether the participants felt it was a true reflection of what they had said [25]. This was done to confirm whether the developed themes were a true representation of the FSWs recounts of their lived experiences with PrEP adherence. Ten of the FSWs were available to review and confirm that the findings were a true reflection of their lived experiences with PrEP adherence. To ensure transferability, an audit trail of all systematic processes used in conducting the study has been kept. The researcher has provided a detailed account of the study and the data was validated by the participants themselves. Bracketing was used to ensure that researcher biases were avoided as far as possible [26–28].

### Ethical considerations

Ethical clearance was obtained from the Biomedical Research Committee, University of KwaZulu-Natal (BREC REF: BREC/00000952/2020), South Africa and the Medical Research Council of Zimbabwe (MRCZ/A/2534) prior to data collection. The FSWs participating in the study were literate, and they had an option to use the English language or their venarcular. The literacy rates in Zimbabwe are higher than other African countries. The FSWs signed the informed consent form confirming their agreement to voluntary participation in the study.

### Findings

As shown in Table 1, twenty FSWs aged between 18 and 55 participated in the study. Four participants had primary school level of education, 11 had secondary school whilst five had completed tertiary education. Fifteen of the FSWs were reportedly divorced, four were single and one reported to be cohabiting with a male partner at the time of the interview. The number of children reported per FSW was between zero and five, with the majority of the participants reporting having three children.

Three main themes and related sub-themes emerged from the data as findings. The three themes were: positive experiences with PrEP adherence, negative experiences with PrEP adherence and the meaning attached to PrEP adherence. All the names used in this section are pseudo-names chosen by the participants themselves.

### Positive experiences with PrEP adherence

This theme emerged from the data and referred to some positive outcomes that came out as a result of FSW's PrEP adherence. The positive experiences were further grouped into four sub-themes which were, overcoming PrEP-related

**Table 1. Demographic characteristics of FSWs on PrEP who participated in the study.**

| Item | Sub-category | N = 20 | Percentage (%) |
|---|---|---|---|
| **Age categories in years** | | | |
| | 18-24 | 4 | 20% |
| | 25-35 | 9 | 45% |
| | 36-45 | 6 | 30% |
| | 46-55 | 1 | 5% |
| **Highest educational level attained** | | | |
| | Primary school | 4 | 20% |
| | Secondary school | 11 | 55% |
| | College level | 5 | 25% |
| **Marital status** | Separated/divorced | 15 | 60% |
| | Never married | 4 | 20% |
| | Cohabiting | 1 | 5% |
| **Number of children participant had** | | | |
| | No children | 2 | 10% |
| | One child | 3 | 15% |
| | Two children | 6 | 30% |
| | Three children | 6 | 30% |
| | Four children | 2 | 10% |
| | More than four children | 1 | 5% |

forgetfulness, overcoming mobility-related PrEP disruptions, overcoming COVID-19 pandemic-related PrEP experiences and overcoming PrEP-related side effects.

**Overcoming PrEP-related forgetfulness**

This sub-theme emerged from data related to participants' struggles with taking HIV-related medication daily when they were not necessarily ill. Participants recounted the associated strategies that they adopted to remind them to take PrEP and address the PrEP adherence challenges as demonstrated below.

*This journey has been very difficult; it took me a long time to get where I am. I used to forget to take my pills all the time. This was not until my friend advised me to set an alarm to remind myself to take my pills every day. Now wherever I am, when I hear my alarm ringing, I know that it's time for my pills (Nokutenda P18, 25–35 years, 3 children).*

*Taking these pills every day has not been an easy experience for me. I used to forget taking my pills all the time. What helped me was my child who is 10 years old and is in grade 3. I told him that I went to the clinic, and I was given pills to take every night at six o'clock. I told him that because I am always forgetful. When my time for my medication is up, he must remind me. My son's daily reminders have been so helpful for me to stay on PrEP. So, if you can find someone close to you whom you can tell so that if you forget or get distracted, that person can remind you. I still remember my child bringing my pills for me whilst at a friend's house (Laugh) (Mukaranga P20, 25–35 years, 3 children).*

Other participants related their struggles and strategies to address forgetfulness as follows:
Overcoming mobility-related PrEP disruptions.

This sub-theme emerged from data on FSWs' experiences of how their mobility and subsequent attachment to clients affected their daily adherence to PrEP. Data analysis revealed that the FSWs who spend a long time away from home

with clients, found themselves with no safe space to store and hide their PrEP. It was noted that some FSWs would join their truck-driver clients who were traveling long distances, sometimes for two weeks to a month, and felt unable to freely take their PrEP, store and or hide it from their clients. Being mobile, with clients, and in someone else's space, meant that FSWs needed to disclose PrEP use to their 'partners' to enable them to continue taking PrEP whilst they were on the road. Some of the FSWs resorted to discontinuing PrEP for the period they were away from home rather than engage their clients in a PrEP conversation which they assumed the clients would not fully comprehend as demonstrated below:

> "There was a time when I traveled and I stopped taking my PrEP. My client at the time did not fully accept there was something called PrEP. He (the client) would not understand the difference between PrEP and ART, so I stopped taking PrEP for that time" (Fortune P2, 36-45 years, 2 children).

> "Sex work is hard; you are with someone for two weeks and you have nowhere to keep your PrEP. You end up just forgetting about PrEP for that time until you are able to go back home" (Judith, P3, 25-35 years, 5 children)

**Overcoming COVID-19 pandemic-related PrEP experiences**

Analyzed data further revealed that some FSWs "got married" to their clients, staying with one client for a number of days, weeks or months within the context of COVID-19. This "marriage" then affected their PrEP use and adherence as indicated in the next quote.

> "During these times of COVID-19, at times I would spend 4-5 days doing sex work and I would not have taken my PrEP with me and then I would not take it (PrEP) until I get back home" (Melo, P13, 25-35 years, 2 children).

> "That is what would make me stop PrEP and then at times I would get into a "marriage" and for you to take PrEP when you are "married" it's a problem. Even if the client is living with HIV, he will be asking you why you are taking PrEP" (Tess P15, 18-24 years, no children).

**Overcoming PrEP-related side effects**

This sub-theme emerged from data related to participants' experiences of PrEP side effects and strategies they developed over time to deal with the PrEP side effects. These strategies enabled FSWS to continue adhering on PrEP as articulated by the following excepts.

> "I used to take my PrEP in the evening, and it would make me feel so weak, nauseated and I would get headaches. This made me fail to go on the streets at night. One of my friends who was already on PrEP noticed that I was not myself, and then advised me to change the time I was taking PrEP. Now I take PrEP in the morning, and it does not affect me that much because I spend most of my day sleeping and only see my clients at night" (Mercy, P12, 25-35 years, 5 children).

> "When I first started on PrEP, I would feel dizzy, nauseated, and would have headaches. I discussed this with my friend who had been on PrEP for much longer and she explained to me that when she started, she would feel the same as well but, as time went on, those side effects eventually went away. She advised me to persist. I persisted on PrEP and within two weeks the side effects were gone. Now, I no longer experience any of those side effects" (Nyarai, P10, 55 years, 4 children).

**Negative experiences associated with PrEP adherence**

This theme emerged from data on the negative experiences associated with PrEP adherence. The theme had two sub-themes namely, enduring GBV and stigma associated with PrEP use and COVID-19-related disruptions to PrEP use.

### Enduring GBV and stigma associated with PrEP use

This sub-theme emerged from data on participants' experiences of ARV stigma. PrEP was usually linked to antiretroviral treatment, with the Female sex workers' clients failing to differentiate between PrEP and ART. This stigma led to FSWs to experience gender-based violence (GBV) such as rape, emotional and physical abuse when their clients discovered that the FSW was using PrEP. The data also revealed that experiences of GBV resulted in some FSWs discontinuing PrEP as reflected in the following quotes:

> "There was a time I met this client who promised to pay me a lot of money. I spent the whole night with him. Now in the morning, I asked for my money, and he started beating me up saying he had seen a bottle of pills in my room. He said I was HIV positive and had given him HIV. This is why I always think of stopping PrEP because it always lands me in trouble with my clients. The client ended up not paying me despite spending the whole night with me" (Paida P8, 25-35 years, 2 children).

> "I once explained PrEP to a client when they saw my bottle of pills in my room. They asked me why I was taking those pills. When I explained that it was PrEP, the client thought I was lying as he thought it was ART. The client ended up accusing me of trying to infect him with HIV. He beat me up badly and he left. I never saw him again" (Mukaranga, P20, 25-35 years, 3 children).

### COVID-19-related disruptions to PrEP adherence

This sub-theme emerged from data on FSWs' experiences of difficulties associated with using and adhering to PrEP during the COVID-19 lockdown. This theme had two sub-themes namely, COVID-19 restrictions affecting access to clients and, limited access to the health facilities due to the COVID-19 lockdowns.

### COVID-19 restrictions affecting access to clients

Female sex workers narrated how the COVID-19-related restrictions affected access to clients thus, reducing the perceived need for PrEP adherence as demonstrated below;

> "COVID-19 has made it difficult to get clients because of the lockdown. So now and again, I would wonder why I should continue taking PrEP when there are no clients. I went for long periods without taking my PrEP. Nothing seemed to work, but then I kept reminding myself why I started taking PrEP in the first place. You also didn't know when you would get a client, so, my PrEP-taking was very inconsistent" (Tsitsi P1, 36-45 years, 3 children).

> "I felt like I was just walking dead, nothing was really working, we had no access to clients due to the COVID-19 lockdown. What is the point of continuing taking PrEP, so I stopped taking PrEP altogether…" (Nyasha P5, 36-45 years, 4 children).

### COVID-19 restrictions affecting access to the clinic

Analyzed data further revealed FSWs were experiencing restricted access to the clinic where they were expected to get PrEP medication, thus affecting their PrEP adherence as illuminated through the following experiences;

> "It was difficult to get to the clinic. You needed a letter to show that you needed the medication" (Chrystal P17, 25-35 years, 1 child).

> "I once forced myself to go to the clinic during the COVID-19 lockdown because I had run out of PrEP and needed it. I got to a police roadblock, and they asked where I was going and what permission I had. I explained that I needed to go

*to the clinic. The police officer told me that only people providing essential services were allowed to move around. My case was not an emergency, and I was told to go back home" (Fortune P6, 36-45 years, 3 children).*

A third theme emerged on the meaning attached to PrEP adherence. This theme emerged from both the negative and the positive experiences with PrEP adherence. Under this theme, there was one sub-theme on commitment to PrEP as a survival strategy.

## PrEP adherence as Survival Strategy

Emerging data revealed that through a combination of both the negative and positive PrEP adherence experiences, some FSWs were forced to remain on PrEP as a survival strategy. This survival strategy enabled the FSWs to remain alive and HIV-negative, despite continuing to sell sex to earn a living hence, the name of the theme. This is demonstrated in the following quotes.

*"Taking PrEP every day has been extremely hard for me…can you imagine I have to take a pill every day like I am living with HIV… but…, as a sex worker I have no choice. I have committed myself to PrEP because I know that my life depends on selling sex. I need to be on PrEP to stay away from HIV and continue to fend for my children through sex work. I have no choice" (Nokutenda, P18, 25-35 years, 3 children).*

*"A lot of the sex workers who are not committed to PrEP, are contracting HIV and eventually dying. For me, taking PrEP has been very difficult but, I have told myself that taking PrEP is the only way I can continue to take care of myself and my children because If you cannot commit to PrEP, you cannot commit to ART as both require commitment to daily medication" (Melo, P13, 25-35 years, 2 children).*

## Discussion and recommendations

The study found three main themes related to experiences with PrEP adherence, including the positive experiences and negative experiences with PrEP adherence and, the meaning attached to PrEP adherence. The findings show that PrEP gave FSWs agency to take control of their own lives as indicated by the measures they took to overcome negative experiences with PrEP adherence. For instance, FSWs experiencing side effects reported having spoken to their peers and receiving encouragement from them, thus underscoring the importance of peer-support and client-centered strategies in enhancing FSW's adherence to PrEP. Similar findings and conclusions have been reported in other studies on PrEP adherence. In Kenya, a pilot study on sex worker outreach program (SWOP) recommended that, engaging young sex workers to identify strategies to enhance retention in FSW programs was critical for higher rates of retention to be achieved. The SWOP study used two main strategies, namely, community outreach and peer education [29]. Another study In Uganda, also recommended the need for peer support among FSWs to help destigmatize PrEP [30]. There were no major variations by demographic characteristics among the FSWs who participated in the study.

The lockdown associated with the COVID-19 pandemic disrupted access to non-essential services. Contrary to literature which shows that sex work actually peaks during periods of strife such as the COVID-19 pandemic, PrEP services were considered a non-essential service and were affected by the COVID-19 lockdown, thus making PrEP adherence almost impossible for some of the FSWs [31–34]. The lack of prioritization of PrEP services as a public health issue, highlights the need for the development and implementation of differentiated delivery of services as proposed by the UNAIDS (2021). Zimbabwe went on to adapt the UNAIDS guidance through it's Operational Services Delivery Manual for the Prevention, Care and Treatment of HIV in Zimbabwe (2022). In their confronting inequalities report, UNAIDS highlights how key populations such as FSWs were left behind in the HIV prevention services as part of the COVID-19 response despite the high burden of HIV. In response to the UNAIDS' implementation of differentiated service delivery, Zimbabwe saw a

marked increase in PrEP initiations during the COVID-19 pandemic when PrEP services were brought to the communities, where the FSWs were [35].

The findings also highlight some negative experiences associated with PrEP adherence, including FSWs reporting experiencing GBV due to their clients discovering that they were on PrEP and, confusing PrEP with ART. Perpetration of GBV against female sex workers who are on PrEP has been reported in South Africa [16,17,19]. This highlights the need for improved community literacy around PrEP, making a clear distinction between PrEP and ART.

The reported FSWs' experiences of PrEP-related side effects negatively affected PrEP adherence in the current study as previously been reported by Busza et al, (2020), Mantsios et al, (2022), Pillay et al, (2020), who all found that fear of side effects was a major deterrent factor for FSWs to continue on PrEP [14,16,18]. We recommend intensive individual counseling on the anticipated PrEP-related side effects for FSWs to be better prepared and increase the chances of PrEP adherence. Additionally, we recommend outreach services for FSWs as a way of bringing the PrEP services to where the FSWs are, and the use of peer support to encourage persistent PrEP use.

### Study strengths and weaknesses

The study sampled the FSWs on PrEP from one clinic. This may bias the findings as the FSWs are likely to have similar experiences. However, the strength of the study is its use of the FSWs' voices and experiences to highlight the psycho-social impact of PrEP Adherence in the lives of FSWs as vulnerable, key populations the qualitative study design enabled the researcher to have a deep understanding of FSW's lived experiences with PrEP adherence in that specific context. This qualitative study has resulted in some rich descriptions of the lived experiences of FSWs with oral PrEP adherence.

### Conclusion

Whilst FSWs reported both positive and negative experiences with PrEP adherence, it is important that FSWs used the meaning they attached to these experiences to take control of their lives and be more determined to use PrEP for survival and protection from HIV infection. Based on these findings, we recommend close monitoring and support to promote adherence, minimize PrEP discontinuity and promote positive lived experiences with PrEP adherence.

### Supporting information

**S1 Data.  FSW lived experiences with PrEP adherence themes.**
(XLSX)

**S1 Questionnaire.  Inclusivity-in-global-research-questionnaire.**
(DOCX)

### Author contributions

**Conceptualization:** Definate Nhamo, Frances M. Cowan, Sinegugu E. Duma.

**Data curation:** Definate Nhamo.

**Formal analysis:** Definate Nhamo.

**Investigation:** Definate Nhamo.

**Methodology:** Definate Nhamo, Frances M. Cowan.

**Project administration:** Definate Nhamo.

**Supervision:** Dixon Chibanda, Frances M. Cowan, Sinegugu E. Duma.

**Writing – original draft:** Definate Nhamo.

**Writing – review & editing:** Definate Nhamo.

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
