## [Decision Letter · Decision Letter 0]

14 Feb 2025

PONE-D-24-43605A Qualitative Analysis of female sex workers’ lived experiences with adherence to Pre-exposure Prophylaxis (PrEP) adherence in Zimbabwe.PLOS ONE

Dear Dr. Nhamo,

Thank you for submitting your manuscript to PLOS ONE. After careful consideration, we feel that it has merit but does not fully meet PLOS ONE’s publication criteria as it currently stands. Therefore, we invite you to submit a revised version of the manuscript that addresses the points raised during the review process.

**ACADEMIC EDITOR:**

Whilst the manuscript provides an important message, the results need to be synthesized and more explanation is needed. The authors present quotes in some sections and insufficient narrative.

The authors should clearly attend to this.

We look forward to receiving your revised manuscript.

Kind regards,

Tinashe Mudzviti, MPhil(MD)

Academic Editor

PLOS ONE

Journal Requirements:

Reviewers' comments:

Reviewer's Responses to Questions

**Comments to the Author**

1. Is the manuscript technically sound, and do the data support the conclusions?

Reviewer #1: Partly

Reviewer #2: Yes

2. Has the statistical analysis been performed appropriately and rigorously? 

Reviewer #1: N/A

Reviewer #2: Yes

3. Have the authors made all data underlying the findings in their manuscript fully available?

Reviewer #1: Yes

Reviewer #2: No

4. Is the manuscript presented in an intelligible fashion and written in standard English?

Reviewer #1: Yes

Reviewer #2: Yes

5. Review Comments to the Author

Reviewer #1: Review of the manuscript entitled: A Qualitative Analysis of female sex workers’ lived experiences with adherence to Preexposure Prophylaxis (PrEP) adherence in Zimbabwe.

Thank you for the opportunity to review the above manuscript. FSWs continue to be at high risk of acquiring HIV. Oral PrEP would certainly help reduce HIV incidence in this population if adherence challenges could be dealt with.

Below are some comments:

Title: Would modifying to include challenges and strategies to improve adherence be better?

The word adherence is repeated.

Abstract:

• Confirm if 10% is prevalence

• What form of PrEP is being referred to? Oral, Injectable or Ring?

Introduction:

• Redefine to Oral PrEP since there are currently various forms of PrEP.

• Also indicate if Truvada (TDF/FTC) since Descovy is available, but approved for MSM

• In the systematic review, what were some of the reasons for discontinuation?

• What are the negative consequences being referred to in the study done in Zimbabwe?

• What would difficulty in understanding PrEP refer to? Understanding the prescription?

• Please clarify the research gap this paper is trying to address as this is not clear from the background provided.

Methods:

• Study design: Was this Cross-sectional or longitudinal? Were any repeat interviews done?

• What was the age group required for the study? What demographics were used for participant selection?

• Were there any refusals during recruitment? What were the reasons?

• Were any notes taken during the interviews?

• Was coding done manually? By how many people? 20 scripts are quite many for one person. Was software used?

• What was the working definition of PrEP adherence?

• Ethical considerations: Were all the FSWs literate?

Results:

• Table 1: Please provide a more comprehensive title for the table

• The characteristic/item for marital status is not provided.

• The quotations provided seem many per sub-theme. Were there any differences per age group or other demographic? Some of the quotes bring out the same points. E.g under forgetfulness- a pill buddy appears twice in the same age group; under side effects, change of time also appears twice in the same age group; most GBV is in the same age group and similar.

• Probably two per theme would suffice

Discussion/conclusion:

• Review the spelling of urgency instead of agency; strife instead of strive?

• What strategies were used in the SWOP study?

• Please confirm that these differentiated service delivery of PrEP is not part of the Zimbabwe guidelines. Many countries adopted these after WHO passed them.

• Under recommendations, what support is being suggested? Please clarify

Reviewer #2: Thank you for submitting this important manuscript that considers the views of the most at risk populations. the manuscript is well structured but requires some minor revisions to make it clear.

Comments:

Introduction:

1. Please state the theoretical orientation that informed your results and describe it applicability.

Results section

2. In the negative theme, you talk about GBV and Stigma. I am wondering whether these are two separate experiences or the same. Please revise and clarify what you are trying to put forward.

3. The narratives in this section are insufficient to guide the reader in understanding how the findings affected the participants.

Discussion:

4. What were the strengths and weaknesses of your study?

6. PLOS authors have the option to publish the peer review history of their article (what does this mean? ). If published, this will include your full peer review and any attached files.

**Do you want your identity to be public for this peer review?** For information about this choice, including consent withdrawal, please see our Privacy Policy .

Reviewer #1: No

Reviewer #2: No

---

## [Author Response · Author response to Decision Letter 1]

10 Apr 2025

Comment Response Pagination

Title:

Reviewer 1

Would modifying to include challenges and strategies to improve adherence be better?

The word adherence is repeated.

We have removed the repetition on adherence to read “A Qualitative Analysis of female sex workers’ lived experiences with adherence to Preexposure Prophylaxis (PrEP) in Zimbabwe”.

P:1

Line 2

Abstract:

Confirm if 10% is prevalence

We confirm that 10% refers to HIV incidence. Page:1

Line 20

What form of PrEP is being referred to? Oral, Injectable or Ring?

We are referring to oral PrEP. This was previously defined from the introduction: “...an HIV prevention option taken as a daily pill”. Page:1

Line 21

Introduction:

Redefine to Oral PrEP since there are currently various forms of PrEP. We defined oral PrEP in the introduction section. The second paragraph starts with defining oral PrEP. Page: 3

Paragraph 2

Line 63 -64

Also indicate if Truvada (TDF/FTC) since Descovy is available, but approved for MSM We have clarified that this is Truvada - TDF/FTC.. We have added the following statement: “The oral PrEP containing Tenofovir - TDF is recommended for use. In Zimbabwe TDF/FTC is preferred, with TDF/3TC as an alternative” Page: 3

Paragraph: 2

Line 66-67

In the systematic review, what were some of the reasons for discontinuation?

We have summarized some of the reported reasons for discontinuation as follows:

“The reasons for PrEP discontinuation were reported as side effects, stigma, influence of partners, difficulty in accessing services, and reduced HIV-risk perception. [

Page: 3

Paragraph: 3

Lines 77-78

What are the negative consequences being referred to in the study done in Zimbabwe? We have clarified the negative consequences as follows: “Female sex workers were faced with negative family discouragement, skepticism, and fear of being given experimental drugs with no proven efficacy”. Page:4

Paragraph: 4

Lines: 85-88

What would difficulty in understanding PrEP refer to? Understanding the prescription?

We have clarified that difficulty in understanding PrEP referred to a lack of complete understanding that oral PrEP needs to be taken during periods of risk. We have replaced the text as follows;

“Female sex workers reported lack of understanding regarding taking and adhering to PrEP as prescribed”. Page:3

Paragraph: 4

Lines: 87-88

Please clarify the research gap this paper is trying to address as this is not clear from the background provided.

The research gap is highlighted in the following text; “This article aims to highlight the FSWs’ psychosocial related experiences on PrEP adherence that could be used to inform the development of a psychosocial intervention – which was the main aim of a bigger study”..

We deleted the following text “this study sought to understand the current lived realities of FSWs with oral PrEP adherence with a view to identifying the key elements to enhance continuation. on page: lines: 113-114

Page: 4

Paragraph: 2

Lines: 113-116

Methods:

Study design: Was this Cross-sectional or longitudinal? Were any repeat interviews done?

This was a cross sectional study with no repeat interviews conducted to understand the lived experiences of PrEP adherence among FSWs on oral PrEP. The following text was added, “These qualitative interviews were conducted once-off, with no repeat interviews held” Page: 5

Paragraph: 2

Lines: 130-131

What was the age group required for the study? What demographics were used for participant selection?

We have added the age and eligibility criteria for qualitative interviews as follows. “To be eligible, one had to be 18 years or older, identify as a female sex worker, be on oral PrEP”

Page: 5

Paragraph: 3

Lines: 136-137

Were there any refusals during recruitment? What were the reasons?

We have added the following text,

“The Female sex workers presenting at the clinic for their first PrEP follow-up visit were informed about the study, and invited the FSWs to participate in the study. The potential participants were given the opportunity to refuse to participate. However, once the study was explained thoroughly, all participants who met the criteria were recruited until data saturation was reached.”

Page: 6

Paragraph: 1

Lines: 148-153

Were any notes taken during the interviews?

Detailed notes were taken during the study, and the interviews were all audio recorded. This is explained as follows: “Detailed notes were taken during the interview. In addition to the notes, all interviews were audio recorded to ensure accuracy in data capturing”.

Page: 7

Paragraph: 1

Lines: 1776-177

Was coding done manually? By how many people? 20 scripts are quite many for one person. Was software used?

The researcher coded the interviews manually with a peer, for intercoder agreement. In addition to this, there was oversight provided by the supervisor who is a qualitative expert. This is explained as follows: “All the coding was done manually by two researchers, the researcher and a fellow qualitative researcher, using an excel sheet. The supervisor, who is a qualitative expert, provided oversight of the coding and analysis process”.

Page: 8

Paragraph: 1

Line: 219-221

What was the working definition of PrEP adherence?

This text has been added: “PrEP adherence was defined as the taking of PrEP during periods of risk. As these were FSWs, the assumption was that their risk was continuous”. Pages: 7

Paragraph: 1

Lines 180-181

Ethical considerations: Were all the FSWs literate?

We have added the following text, “The FSWs participating in the study were literate, and they had an option to use the English language or their venarcular. The literacy rates in Zimbabwe are higher than other African countries ”. Pages: 9

Paragraph: 2

Lines: 248-251

Results:

Table 1: Please provide a more comprehensive title for the table

The following text has been added “Demographic Characteristics of FSWs on PrEP who participated in the study”

Pages: 9

Paragraph: 4

Lines: 262-263

The characteristic/item for marital status is not provided.

We have added the item for “marital status” Pages: 10

Paragraph: 4

Lines: 263

The quotations provided seem many per sub-theme. Were there any differences per age group or other demographic? Some of the quotes bring out the same points. E.g under forgetfulness- a pill buddy appears twice in the same age group; under side effects, change of time also appears twice in the same age group; most GBV is in the same age group and similar.

We note the observation. However, the intention was not to compare themes or experiences per age group as done by the reviewer, but to analyse and identify data that has similar meaning for the development of appropriate themes.

Probably two per theme would suffice

We have reduced the quotes to two per theme as per the reviewer’s advice.

Discussion/conclusion:

Review the spelling of urgency instead of agency; strife instead of strive? We have reviewed the spellings of “agency’, which meant, the FSWs had a sense of “control”. We have left it as it was.

We have also reviewed the spelling of “strive”, and replaced it with “strife”, as this was referring to periods of difficulty, the FSWs were experiencing during COVID 19. Pages: 17

Paragraph: 2

Lines: 469

What strategies were used in the SWOP study?

The SWOP study applied strategies around the establishment and implementation of effective outreach and peer education plans. We have added the following text, “The SWOP study used two main strategies, namely, community outreach and peer education”.

Pages: 17

Paragraph: 1

Lines: 461- 462

Please confirm that these differentiated service delivery of PrEP is not part of the Zimbabwe guidelines. Many countries adopted these after WHO passed them. These differentiated service delivery of PrEP are part of the Zimbabwe guidelinese adapted from the UNAIDS guidelines We have added the following text; “Zimbabwe went on to adapt the UNAIDS guidance through it’s Operational Services Delivery Manual for the Prevention, Care and Treatment of HIV in Zimbabwe (2022)”. Pages: 17

Paragraph: 2

Lines: 475-477

Under recommendations, what support is being suggested? Please clarify

We have added the following text to clarify the support we are suggesting. “Additionally, we recommend outreach services for FSWs as a way of bringing the PrEP services to where the FSWs are, and the use of peer support to encourage persistent PrEP use”.

Pages: 18

Paragraph: 2

Lines: 497-499

Reviewer 2

Introduction:

Please state the theoretical orientation that informed your results and describe it applicability.

The findings were informed by a feminist lens. Page: 8

Paragraph: 1

Results:

Lines: 225

In the negative theme, you talk about GBV and Stigma. I am wondering whether these are two separate experiences or the same. Please revise and clarify what you are trying to put forward.

We note the observation. However, in our analysis, the stigma was associated with taking PrEP. This stigma could be from partners or any other community members. GBV, in particular was related to violence against FSWs once discovered to be using “ARVs” thus exposing the partner to the virus.

The narratives in this section are insufficient to guide the reader in understanding how the findings affected the participants.

The direct quotations help bring to life the FSW experiences from their lenses and not from the researcher’s lens. We hope this helps give some vivid descriptions of the untold stories and experiences of FSWs on oral PrEP.

Discussion:

What were the strengths and weaknesses of your study?

We have added a new section on the study strengths and weaknesses. Pages: 18

Paragraph: 2

Lines: 501-509

---

## [Decision Letter · Decision Letter 1]

29 Jul 2025

A Qualitative Analysis of female sex workers’ lived experiences with adherence to Pre-exposure Prophylaxis (PrEP)  in Zimbabwe.

PONE-D-24-43605R1

Dear Dr. Nhamo,

We’re pleased to inform you that your manuscript has been judged scientifically suitable for publication and will be formally accepted for publication once it meets all outstanding technical requirements.

Kind regards,

George Vousden

Deputy Editor in Chief

PLOS ONE 

Additional Editor Comments (optional):

Reviewers' comments:

Reviewer's Responses to Questions

**Comments to the Author**

1. If the authors have adequately addressed your comments raised in a previous round of review and you feel that this manuscript is now acceptable for publication, you may indicate that here to bypass the “Comments to the Author” section, enter your conflict of interest statement in the “Confidential to Editor” section, and submit your "Accept" recommendation.

Reviewer #1: All comments have been addressed

2. Is the manuscript technically sound, and do the data support the conclusions?

Reviewer #1: Yes

3. Has the statistical analysis been performed appropriately and rigorously? 

Reviewer #1: Yes

4. Have the authors made all data underlying the findings in their manuscript fully available?

Reviewer #1: Yes

5. Is the manuscript presented in an intelligible fashion and written in standard English?

Reviewer #1: Yes

6. Review Comments to the Author

Reviewer #1: Thank you for thoroughly addressing all of the comments.

Review certain sections to ensure proper grammar and readability.

7. PLOS authors have the option to publish the peer review history of their article (what does this mean? ). If published, this will include your full peer review and any attached files.

**Do you want your identity to be public for this peer review?** For information about this choice, including consent withdrawal, please see our Privacy Policy .

Reviewer #1: No

---

## [Editor Report · Acceptance letter]

PONE-D-24-43605R1

PLOS ONE

Dear Dr. Nhamo,

I'm pleased to inform you that your manuscript has been deemed suitable for publication in PLOS ONE. Congratulations! Your manuscript is now being handed over to our production team.

Kind regards,

on behalf of

Dr. George Vousden

Staff Editor

PLOS ONE